# 3D-Printed Phenylboronic Acid-Bearing Hydrogels for Glucose-Triggered Drug Release

**DOI:** 10.3390/polym16172502

**Published:** 2024-09-03

**Authors:** Jérémy Odent, Nicolas Baleine, Serena Maria Torcasio, Sarah Gautier, Olivier Coulembier, Jean-Marie Raquez

**Affiliations:** Laboratory of Polymeric and Composite Materials (LPCM), Center of Innovation and Research in Materials and Polymers (CIRMAP), University of Mons (UMONS), Place du Parc 20, 7000 Mons, Belgium

**Keywords:** 3D printing, phenylboronic acid, hydrogels, drug-eluting implants

## Abstract

Diabetes is a major health concern that the next-generation of on-demand insulin releasing implants may overcome via personalized therapy. Therein, 3D-printed phenylboronic acid-containing implants with on-demand glucose-triggered drug release abilities are produced using high resolution stereolithography technology. To that end, the methacrylation of phenylboronic acid is targeted following a two-step reaction. The resulting photocurable phenylboronic acid derivative is accordingly incorporated within bioinert polyhydroxyethyl methacrylate-based hydrogels at varying loadings. The end result is a sub-centimeter scaled 3D-printed bioinert implant that can be remotely activated with 1,2-diols and 1,3-diols such as glucose for on-demand drug administration such as insulin. As a proof of concept, varying glucose concentration from hypoglycemic to hyperglycemic levels readily allow the release of pinacol, i.e., a 1,2-diol-containing model molecule, at respectively low and high rates. In addition, the results demonstrated that adjusting the geometry and size of the 3D-printed part is a simple and suitable method for tailoring the release behavior and dosage.

## 1. Introduction

Diabetes is a chronic disease characterized by persistently high blood glucose levels, making it one of the most prevalent global causes of mortality and morbidity [1]. The current therapy is provided by insulin administration through invasive delivery methods such as repetitive subcutaneous injections [2]. This mode of administration is painful and may lead to injection phobia, infections and incorrect medication dose [3]. In this regard, significant efforts have been made to develop long-lasting insulin delivery systems [4,5]. Since the in vivo insulin fluctuation is not linear, the elaboration of such systems requires the modulation of insulin release through the action of biological stimuli such as the glucose concentration [6]. Therein, insulin–lectin (e.g., concanavalin A) complexes are designed to release insulin upon a competitive binding with the triggering glucose, but its competitive set-point is above the range of typical hyperglycemic concentrations [7,8,9]. Alternatively, chemically controlled glucose responsive systems have been widely investigated over the last few decades [10]. Among them, pH-responsive materials allow on-demand release of a pre-loaded insulin dose using a glucose-specific enzyme (e.g., glucose oxidase) that can convert the glucose into gluconic acid in the presence of oxygen, thereby causing a decrease in local pH [11,12,13]. Despite high activity and low toxicity, these systems suffer from major hurdles, particularly, the loss of biological activity after oral administration, their blood oxygen dependence and their high sensitivity to environmental changes [14,15,16]. Alternative glucose-responsive systems based on phenylboronic acid (PBA), which can program the insulin release rate in response to different glucose concentrations, have therefore emerged [10,17]. PBA is likely well-known to have substantial affinities for glucose because of its ability to bind specifically and reversibly with 1,2- and 1,3-diols through reversible boronate ester formation [18,19,20,21,22,23]. In addition, PBA has high operational stability, in particular when compared with glucose oxidase and lectin, which are widely used for glucose sensing [24].

Fast and stable bond formation between PBA and glucose enable reversible molecular assemblies capable of trapping and releasing insulin for the tight regulation of blood glucose levels [25,26,27,28,29,30]. It is therein assumed that PBA–glucose complexes of both 2:1 and 1:1 stoichiometries are formed [31]. As the glucose concentration increases, the 2:1 complex is thereby converted to two 1:1 complexes, leading to a decrease in the number of crosslinks and an increase in the number of free anionic 1:1 complexes, which in turn entail an expansion in the hydrogel volume [31]. The resulting glucose-dependent volumetric changes ultimately accelerate the release kinetics of physically entrapped insulin. In an effort to delineate the key parameters influencing the PBA–glucose esterification, Zhang et al. demonstrated that substituents in the aromatic ring of PBAs further affect the kinetics of the esterification/transesterification [32]. In addition, Achilli et al. evidenced that the hydrolysis of boronate esters is dependent on both the substituents in the aromatic ring and the pH of the solution [33]. Although, the release kinetics can be controlled remotely through molecular design, novel manufacturing technologies surely offer new possibilities to maximize personalization through the combination of the right materials with the right three-dimensional geometry.

Healthcare innovations to medical devices and manufacturing will enable devices to be more modular and adaptable while creating new opportunities to improve the personalization, quality and safety of treatment. Additive manufacturing (AM), i.e., 3D printing, has become one of the most innovative technologies in the pharmaceutical field [34,35]. Currently, multiple 3D printing techniques and materials are investigated with regard to manifold characteristics for drug-eluting implants such as shape, surface, microstructure, mechanical properties and drug release behavior [36]. Therein, complex microstructures such as internal channeled [37,38], grid-like [39] or honeycomb [40] already demonstrated their ability to tailor drug release. Stereolithography (SLA) offers the highest versatility with respect to freedom in material development, scalability and speed of fabrication [41,42,43]. The advantages of SLA compared to all other AM, when biological (macro)molecules such as proteins [44,45,46], genetic materials (e.g., DNA, RNA) [47,48,49] and other biological molecules [32,50,51] are concerned, have been well-established thanks to its mild operating conditions that prevent undesired degradation [52]. The development of biocompatible materials that enable the fabrication of large-scale 3D-printed objects are still required [53,54]. Therein, our group has investigated the multivariate photochemistry of SLA and reported the rapid 3D printing of tough, highly solvated and antifouling hybrid zwitterionic hydrogels for potential uses in biomedical applications [55]. In addition, bioinert composite elastomers that can be rapidly 3D-printed into complex geometries with high-resolution features via SLA 3D printing that are expected to contribute to the development of medical devices from readily available materials are produced [56,57]. Interestingly, Robinson et al. exploited boronate esters as a key building block for the development of 3D-printed objects with the ability to undergo room-temperature exchange at the crosslinking sites [58]. As far as insulin delivery systems are concerned, Wu et al. 3D-printed insulin-loaded PBA-containing microneedle patches, allowing the diffusive release of entrapped insulin [59]. However, 3D-printed tailored dosage forms with on-demand glucose-triggered drug release abilities are still needed.

In the present work, new bioinert PBA-containing polyhydroxyethyl methacrylate (PHEMA)-based hydrogels of tailored dosage forms are 3D-printed at high-resolution features using commercial SLA technology. Methacrylation of phenylboronic acid is thereby envisioned for its effective incorporation within bioinert PHEMA-based hydrogels at varying loadings. The resulting glucose-responsive systems should thereby endow reversible volume change with respect to the glucose concentration as the 2:1 complex is converted to two 1:1 complexes, and vice versa. Pinacol, a 1,2-diol-containing model molecule, is used to demonstrate the controlled glucose-triggered drug release ability of our system [60]. As far as insulin administration is concerned, diol-functionalized insulin is required instead of pinacol to endow the precise regulation of blood glucose level. Therein, Hashimoto et al. already demonstrated that the biological activity of insulin is maintained as high as ca. 90% after its chemical modification [61]. Overall, the present contribution further demonstrated our ability to photo-pattern complex 3D objects with unusual geometries on demand to comply with the individual needs of each patient in term of release and dosage forms.

## 2. Experimental

### 2.1. Materials

4-Carboxybenzeneboronic acid (4CPBA, ≥99%, Apollo Scientific, Stockport, UK), 2-hydroxyethyl methacrylate (HEMA, 97%, Merck, Darmstadt, Germany), poly(ethylene glycol) methacrylate (PEGMA, Mw = 500 g/mol, ≥99%, Sigma-Aldrich, Saint Louis, MO, USA), poly(ethylene glycol) dimethacrylate (PEGDMA, Mw = 550 g/mol, ≥99%, Sigma-Aldrich, Saint Louis, MO, USA), 2,3-dimethylbutane-2,3-diol (pinacol, ≥99%, Merck, Darmstadt, Germany), diphenyl(2,4,6-trimethylbenzoyl) phosphine oxide (TPO, 97%, Sigma-Aldrich, Saint Louis, MO, USA) and trifluoroacetic anhydride (TFAA ≥ 99%, Sigma-Aldrich, Saint Louis, MO, USA) are purchased as indicated and used without further purification. Tetrahydrofuran (THF, 99%, VWR, Leuven, Belgium) is dried using a MBraun Solvent Purification System (model MB-SPS 800, MBraun, München, Germany) equipped with alumina drying columns.

### 2.2. Synthesis of Phenylboronic Acid-Based Methacrylate

The reaction involves a two-step synthesis. The first step relies on an equimolar reaction of 4CPBA (m = 500 mg, n = 3 mmol) with pinacol (m = 356 mg, n = 3 mmol) in 10 mL of dry THF. After 3 h under agitation at room temperature, the as-protected phenylboronic acid is isolated under vacuum for 12 h, leading to compound **1** (see Figure 1A). In a second step, the resulting pinacol protected 4-carboxybenzeneboronic acid (m = 747.6 mg, n = 3 mmol) and 1 eq. of TFAA (m = 623.8 mg, n = 3 mmol) are mixed to 10 mL of dried THF at room temperature. After 15 min, 1 equivalent of PEGMA (m = 1.51 g, n = 3 mmol) is slowly added to the reaction medium. The solution is allowed to stir for an additional 12 h at room temperature. The resulting α-methacrylate ω-pinacolyl boronic ester poly(ethylene glycol) (PEGMA-PBA-Pinacol), i.e., compound **2** (see Figure 1A), is isolated by simple drying procedure under vacuum. Yield = 98%, M_n_ = 788.53 g/mol, Ɖ_M_ = 1.16, ^1^H-NMR (DMSO-*d*_6_, 500 MHz): δ(ppm): 7.95 (2H, CH), 7.79 (2H, CH), 6.03–5.69 (2H, CH_2_), 4.51(2H, CH_2_-0-CO), 4.21 (2H, CH_2_-O), 1.88 (3H, CH_3_-CO), 1.31 (12H, CH_3_); FT-IR (ATR, cm^−1^): 1358 (B-O).

### 2.3. 3D Printing of Bioinert Phenylboronic Acid-Containing Hydrogels

Different ratios in HEMA and compound **2** are used in conjunction with 1 mol% of PEGDMA and 0.5 mol% of TPO. A full-spectrum light exposure (OmniCure S1000, Lumen Dynamics, Omnicure Saint Louis, MO, USA, 405 nm) allows the photocuring of the formulations in several shapes (e.g., disks). The 3D printer is a Liquid Crystal Precision 1.5 by Photocentric using an LCD screen of 405 nm wavelength, controlled by an open-source software. All the .stl files are designed using Autodesk Fusion 360 v.2.0.18719 modeling software and then imported on Photocentric studio to slice the design into discrete 100 µm layers. Layer exposure times are set at 45 s per discrete 100 µm layer. Resulting 3D objects (capsules of 30 mm length and 10 mm diameter) are washed successively in an excess of isopropanol for 24 h and in a PBS solution for an extra 24 h. While the gel fraction is systematically over 97%, no potentially toxic unreacted moieties are released from the materials, as confirmed by ^1^H-NMR after washing. Samples are named PHEMA-PBAx with x corresponding to the molar percentage of PEGMA-PBA-Pinacol (i.e., compound **2**) in the hydrogel matrix.

### 2.4. Transesterification of Phenylboronic Pinacol Ester with Glucose

The stability of compound **2** towards glucose transesterification is evaluated in a 100 mM sodium phosphate buffer at pH 7.4. To that end, compound **2** is solubilized in three buffer solutions at an initial concentration of ca. 4 mM containing different amounts of glucose (i.e., 4, 8 and 12 mM). Phenylboronic pinacol ester–glucose transesterification is quantified by ^1^H-NMR spectroscopy. We can further determine the nature of diffusion of the entrapped drug from the hydrogels by plotting the relative pinacol release vs. the square root of immersion time. The diffusion coefficients (*D*) could be further estimated from the slope-values according to Equation (1).
(1)$$\varphi =\frac{4}{d}\sqrt{\frac{Dt}{\pi }}$$
where *D* is the diffusion coefficient, *φ* is the relative pinacol release and *d* is the thickness of the sample in cm.

### 2.5. In Vitro Drug Release Testing

Samples are immersed in a pH 7.4 phosphate buffer saline (PBS) solution with various concentrations of glucose (i.e., 0, 4, 8 and 12 mM). A total of 5 mL of the buffer solution is taken out at preset time points and samples are re-immersed in fresh buffer solution. Pinacol release is followed by ^1^H-NMR spectroscopy. Drug release testing is performed on 5 specimens with identical parameters to provide mean values and standard deviations.

### 2.6. Additional Techniques

^1^H-NMR spectra are recorded on a Bruker Avance III HD 500 MHz spectrometer at 293 K. Chemical shifts are reported as δ in parts per million (ppm) and referenced to the residual solvent signal (DMSO-*d*_6_, δ = 2.50 ppm).

Fourier transform infrared spectroscopy (FTIR) measurements are carried out with a Spectrum One FTIR spectrometer (Perkin Elmer Instruments, Shelton, CT, USA). Spectra are obtained in transmission mode at room temperature in the 600–4000 cm^−1^ region with a resolution of 4 cm^−1^ and 32 scans.

Photo-rheological measurements are performed using a rheometer MCR 302 (Anton-Paar) connected to a UV-light source (OmniCure S1000, Lumen Dynamics). During photo-exposure (irradiance E_e_ = 22.5 mW/cm^2^), frequency (ω = 1 Hz) and amplitude (ε = 1%) oscillatory shear are constant. Samples are placed between circular parallel plates (diameter = 25 mm) with a gap of 1 mm.

Rheological measurements are performed using a rheometer MCR 302 (Anton-Paar, Graz, Austria) using a plate–plate geometry system with a 25 mm diameter and a gap of 1 mm. Strain sweep measurements are performed at 25 °C with a frequency of 1 Hz and a strain range between 0.01% and 100%. All samples are swelled at equilibrium in PBS solution containing different glucose concentrations before measurement (i.e., from 0 to 12 mM). The relationship between the storage modulus in the rubbery plateau and the crosslinking density is given by the following equation:(2)$$\vartheta_{e}=\frac{G^{\prime }}{RT}$$
where *ν_e_* is the crosslinking density, *G*′ is the shear modulus obtained in the rubbery plateau, *R* is gas constant and *T* is the temperature in Kelvin. The relationship between the storage modulus in the rubbery plateau and the mesh size is given by the following equation:(3)$$r_{mesh}=\sqrt[{3}]{\frac{6RT}{\pi N_{Av}G^{\prime }}}$$
where *r_mesh_* is the mesh size, *R* is the gas constant, *T* is the temperature in Kelvin, *N_Av_* is Avogadro’s number and *G*′ is the shear modulus in the rubbery plateau region.

## 3. Results and Discussion

### 3.1. 3D-Printed Bioinert Phenylboronic Acid-Containing Hydrogels

The number of commercially available resins suitable for SLA 3D printing being limited, the approach involved the end-functionalization of PBA, as a glucose-responsive trigger, with highly reactive methacrylate groups to readily undergo rapid free radical photo-polymerization under near-UV exposure. The synthesis of photocurable PBA-based glucose-responsive derivative is performed following a two-step reaction (Figure 1A). First, treatment of 4CPBA with pinacol afforded the corresponding pinacolyl boronate esters, i.e., compound **1** (see Figure 1A). Unlike its boronic acid precursor being more prone to dehydration to afford an insoluble anhydride trimeric boroxine [62], 4CPBA is remarkably soluble in common organic solvents, allowing its reaction to be conducted in dried THF at room temperature over 3 h. The resulting pinacolyl boronate ester is then esterified with a low molecular weight PEGMA under very mild conditions using TFAA, which is converted into the corresponding α-methacrylate ω-pinacolyl boronic ester poly(ethylene glycol), i.e., compound **2** (see Figure 1A) [63]. Such an approach obviates the necessity of a multi-step process while enabling a direct reaction to occur between the acidic group of 4CPBA and the hydroxyl function of PEGMA at room temperature over 12 h. ^1^H-NMR analyses clearly evidenced the quantitative formation of compound **1** and compound **2** (Appendix A in the Appendix A). As the transesterification reaction of phenylboronic pinacol esters is known to be dependent on both the substituents in the aromatic ring and the pH of the solution, the second step of this synthesis readily allows the attachment of an electron-donor PEG segment in para position to the boronic ester derivative, thus reducing undesired drug release. Indeed, Achilli et al. demonstrated that the rate of hydrolysis is slower when an electron-donor group is attached in para position to the boronic moiety on the aromatic ring [33]. Moreover, it allows the attachment of a photopolymerizable methacrylate moiety to the boronic ester derivative, thus allowing its use for SLA 3D printing.

The stability of compound **2** towards competitive transesterification of the boronate with glucose is first evaluated in a 100 mM sodium phosphate buffer at pH 7.4. To mimic conditions associated to hypo-, normo- and hyperglycemic levels, the conversion of phenyl boronic pinacol esters to phenyl boronic glucose ester by transesterification reaction at varying glucose concentrations (i.e., 4, 8 and 12 mM, respectively) is determined after three days (i.e., at thermodynamic equilibrium) by ^1^H-NMR (Appendix A). In hypoglycemic conditions (i.e., 4 mM), only ca. 31% of pinacol is released, suggesting very little of the transesterification reaction between the free diols and boronate ester bonds (Appendix A). As the glucose concentration increases, the transesterification reaction turns favorable, releasing ca. 41% and ca. 84% of pinacol at normoglycemic (i.e., 8 mM) and hyperglycemic (i.e., 12 mM) conditions, respectively (Appendix A). Note that the nearly complete transesterification in hyperglycemic conditions readily allows the effective release of entrapped cargo for on-demand drug administration such as insulin.

Photo-rheology allows for the in situ monitoring of chemical and mechanical characteristics during the photo-polymerization reactions, providing key data concerning the suitability of the as-synthesized compound **2** towards SLA 3D printing technology (Appendix A) [64,65]. The key requirement is the ability to rapidly reach a high degree of conversion using a near-UV source of equivalent wavelength to our SLA equipment. Compound **2** is thereby incorporated within PHEMA-based hydrogels at varying concentrations (i.e., 2–10 mol%). While PEGDMA crosslinker is used at 1 mol%, TPO is used as a type-I photo-initiator, enabling free radical photo-polymerization of the acrylate groups with near-UV SLA systems (Figure 1B). Initially exhibiting a liquid-like behavior with a loss modulus (G″) greater than the storage modulus (G′) and with an initial viscosity below 1 Pa.s, the viscosity starts to increase as the gelation process occurs under near-UV exposure, leading to solid-like behavior as characterized by G′ greater than G″. Resulting compound **2** displayed slower polymerization kinetics than neat HEMA with a gel point occurring at longer near-UV exposures (ca. 38 s vs. 32 s, see Appendix A). When combined, the new material thereby displays an intermediate behavior (i.e., 34 s at 10 mol% of compound **2** within HEMA). HEMA and compound **2** (or combination) are, however, suitable for SLA 3D printing, allowing us to design PBA-containing PHEMA-based hydrogels (see Figure 1B). Fabrication of such hydrogels into complex geometries of high-resolution features is performed on an LC Precision 1.5 (Photocentric, λ = 405 nm) SLA printer. The characteristic B-O asymmetric stretching band at 1358 cm^−1^ in the FTIR spectra confirmed the incorporation of boronic acid moieties within the PHEMA-based hydrogels (Appendix A).

### 3.2. Glucose Responsiveness Features

The unique feature of boronic acid is that it is capable of forming reversible covalent complexes with 1,2- or 1,3-diols, including glucose [31]. While boronate ester formation leads to the creation of a 2:1 complex at low glucose concentrations, the later 2:1 complex is converted to two 1:1 complexes as the glucose concentration increases (Figure 2A). As a result, a decrease in the number of crosslinks and an increase in the hydrogel volume, which in turn accelerates the release kinetics of physically entrapped drugs, including insulin, are expected. In hypoglycemic conditions (i.e., 4 mM), PBA is largely in excess compared to glucose, leading to 2:1 complexes and a highly crosslinked PHEMA-based hydrogel network (Figure 2B). In contrast, glucose saturates boronic acid sites as the glucose concentration increases over a critical limit (i.e., beyond normal glucose conditions, >8 mM), converting a 2:1 complex into 1:1 complexes with respect to the loosely crosslinked PHEMA-based hydrogel network (Figure 2B). The later conversion at varying glucose concentrations from hypoglycemic to hyperglycemic levels (i.e., from 4 to 12 mM) is evidenced by rheological measurements (Appendix A). Oscillation amplitude sweeps thereby attested for a typical elastic solid behavior since storage modulus (G′) in each case is higher than loss modulus (G″). An estimation of the effective crosslinking density is further assessed by storage modulus measurements in the rubbery plateau region given Equation (2) [66]. In addition, the mesh sizes which refer to average distance between the entanglements within the resulting PHEMA-based hydrogel network are further estimated by Equation (3) [67]. While the estimated crosslinking density and related mesh size of the neat PHEMA hydrogel does not change at varying glucose concentrations, PBA-containing PHEMA-based hydrogels show a significant decrease in the estimated crosslinking density and increase of the mesh size from hypoglycemia to hyperglycemia (see Appendix A). In addition, the higher the PBA loading within the PHEMA-based hydrogel, the more crosslinked hydrogel networks with respect to higher crosslinking density and lower mesh size. Recall that highly crosslinked 2:1 complexes are formed in hypoglycemic conditions which are converted into loosely crosslinked 1:1 complexes in hyperglycemic conditions. Such a decrease in the number of crosslinks and increase in the mesh size thereby engender an expansion in the hydrogel volume which should ultimately accelerate the release kinetics of the physically entrapped drug, including insulin. While the drug release is mainly dominated by a diffusion process, if the mesh size is larger than the hydrodynamic radius of the diffusive drug molecule, similar or smaller mesh size than the hydrodynamic radius of the diffusive drug molecule will result to a slower drug release rate with respect to steric hindrance. Note that insulin is either found in the body as a free molecule or as a complexed hexamer of ca. 3 nm for storage purposes [68]; thus, diffusion of physically entrapped insulin should be dominated by steric hindrance in hypoglycemic and normal conditions with respect to recorded mesh sizes around ca. 3 nm, while rather dictated by diffusion process in hyperglycemic conditions with respect to mesh sizes around ca. 5 nm.

### 3.3. Drug Release Kinetics

Pinacol, a model molecule, is used to demonstrate the controlled glucose-responsive drug delivery ability of the resulting 3D-printed PHEMA-based hydrogels. The main difference between pinacol and insulin release would be attributed to steric hindrance, which results in a slower release rate for insulin [69]. Note, though, that diol-functionalized insulin is required to ensure the precise regulation of the blood glucose level [61]. To study the release kinetics, PHEMA-PBAx (with x = 2 or 10 mol%) is first immersed in buffer solution in the absence of glucose (i.e., 0 mM) to determine the background rate of release due to hydrolysis of the pinacol ester (Figure 3). Overall, a pinacol release as low as ca. 1% after 3 days attested for very little hydrolysis. Subsequently, the amount of released pinacol is measured by ^1^H-NMR analyses over time (i.e., up to 72 h) at varying glucose conditions, going through hypo-, normo- and hyperglycemia (Figure 3). As a result, a rapid release of pinacol is observed during the first hour, suggesting a burst effect, as evidenced by the perfect fit of the rapid release period with the fast swelling of the hydrogel (Appendix A) [70]. The burst effect is followed by a steady release. While the glucose concentration does not seem to impact the burst release with an effective release of ca. 15% of pinacol after 1 h for all compositions, the steady release is clearly affected under varying glucose levels. While PHEMA-PBA10 releases ca. 30–33% of pinacol after 24 h (and ca. 45–50% after 3 days) with a steady release of ca. 0.45%/h (i.e., 1 × 10^−6^ mol/h) in hypoglycemic (i.e., 4 mM) and normoglycemic (i.e., 8 mM) conditions, the release is boosted to ca. 49% (and 84% after 3 days) under hyperglycemic (i.e., 12 mM) conditions (see Figure 3). The pinacol release as a function of the square root of time show a linear increase for the steady release period, suggesting simple Fickian diffusion (Appendix A), leading to the diffusion coefficient derived from Equation (3) (Appendix A) [71]. While the diffusion coefficient does not seem to vary with the chemical incorporation of PBA (i.e., via compound 2) within the PHEMA-based hydrogel, the diffusion coefficient increases from hypoglycemic to hyperglycemic levels. Recall that the estimated mesh size of the resulting PHEMA-based hydrogel networks increases with respect to the glucose concentration, leading to an expansion in the hydrogel volume which ultimately accelerates the release kinetics of the physically entrapped drugs (see Appendix A). The high sensitivity toward glucose is further assessed by alternatively immersing PHEMA-PBA10, under hypoglycemic and hyperglycemic conditions every 30 min (Figure 4). Overall, high release of pinacol is observed under hyperglycemic conditions (i.e., 12 mM), while low release (remaining almost constant) is recorded under hypoglycemic condition (i.e., 4 mM). Overall, ca. 1% of pinacol is released during the first cycle, to equilibrate around ca. 0.5% during the second and third cycles. The latter results suggest that PBA-containing PHEMA-based hydrogels can be used as an on–off switching system as a function of the glucose concentration, enabling on-demand release of a desired molecule. To demonstrate function, resulting glucose-responsive PHEMA-based hydrogels are 3D-printed into centimeter-scaled drug eluting implants, i.e., capsules of 30 mm length and 10 mm diameter, with modifications in the geometry and size resulting in changes to the drug release and dosage behavior. Therein, adjusting the specific surface-area-to-volume ratio of the drug eluting implants by controlling the 3D printing infill density from 100% (i.e., solid) to 50% (i.e., opened structure) appears to be a simple way to endow faster drug release kinetics (Figure 5A) [39]. As far as the pinacol release is concerned, the drug eluting implant with an infill density of 50% provided a faster burst effect as well as a higher steady release than the implant with an infill density of 100% (Figure 5B and Appendix A). While the implant with an infill density of 50% showed a burst effect during the first 40 min with a release of ca. 15%, the implant with an infill density of 100% lasts one hour to reach ca. 15% of release. In addition, release rates of ca. 40%/day and ca. 22%/day, leading to a complete release after 48 h and 72 h, are, respectively, recorded for the implants with an infill density of 50% and 100% in hyperglycemic conditions. The difference in the pinacol release is further assessed by plotting the amount of released pinacol as a function of time (Figure 5C). Therein, a release rate of ca. 4.1 × 10^−5^ mol/day and ca. 5.4 × 10^−5^ mol/day in hyperglycemic conditions is observed for the implants with an infill density of 50% and 100%, respectively. If we rationalize this rate as a function of the volume of the implant, the release rate becomes 1.7 × 10^−5^ mol.mm^−^^3^.day and 1 × 10^−5^ mol.mm^−^^3^.day, respectively, for the implants with an infill density of 50% and 100%. Recall that the lower the infill density, the higher the surface-area-to-volume ratio, which, in turn, allows faster release of entrapped drug. The latter demonstrated the ability to tailor drug release through the combination of the right material with the right three-dimensional geometry using high resolution stereolithography technology for personalized therapy.

## 4. Conclusions

3D-printed PBA-containing PHEMA-based hydrogels with on-demand glucose-triggered drug release abilities are successfully produced using high resolution SLA technology as the next-generation of drug eluting implants for personalized therapy. To that end, photocurable PBA-based glucose-responsive derivatives are synthesized following a two-step reaction. The resulting derivatives are incorporated within bioinert PHEMA-based hydrogels at varying loadings, leading to centimeter-scaled 3D-printed drug eluting implants that can be remotely activated with diols such as glucose for on-demand drug administration such as insulin. Resulting glucose-responsive systems endow reversible volume change with respect to the glucose concentration as the 2:1 complex is converted to two 1:1 complexes, leading to a decrease of the number of crosslinks, which ultimately accelerates the release kinetics of the physically entrapped drug. In the present work, pinacol, a 1,2-diol-containing model molecule, is used to demonstrate the controlled glucose-triggered drug release ability of our system. A 2 × 10^−6^ mol/h released rate has been observed under hyperglycemic conditions. However, diol-functionalized insulin is required instead of pinacol to endow the precise regulation of the blood glucose level. To demonstrate function, 3D-printed centimeter-scaled drug eluting implants are fabricated using high resolution SLA technology. Adjusting the specific surface-area-to-volume ratio of the resulting implants by controlling the 3D printing infill density further demonstrated the ability to tailor drug release through the combination of the right material with the right three-dimensional geometry for personalized therapy.

## Figures and Tables

**Figure 1 polymers-16-02502-f001:**
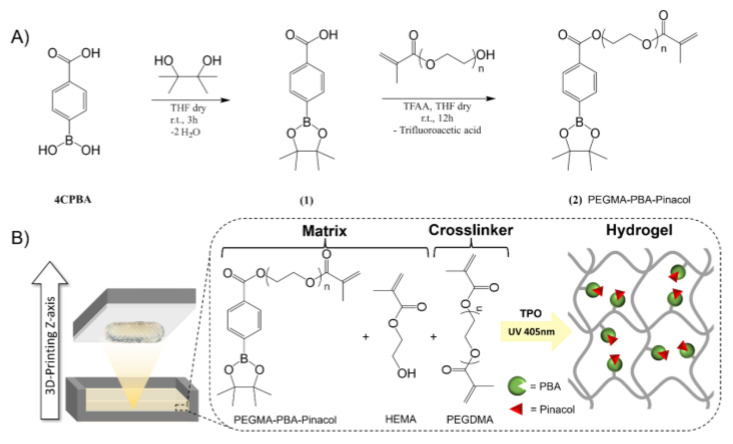
(**A**) Synthetic route of α-methacrylate ω-pinacolyl boronic ester poly(ethylene glycol) (PEGMA-PBA-Pinacol, i.e., compound **2**). (**B**) Fast bottom-up fabrication of PBA-containing hydrogels via photopolymerization of HEMA and PEGMA-PBA-Pinacol using PEGDMA crosslinker and TPO photo-initiator.

**Figure 2 polymers-16-02502-f002:**
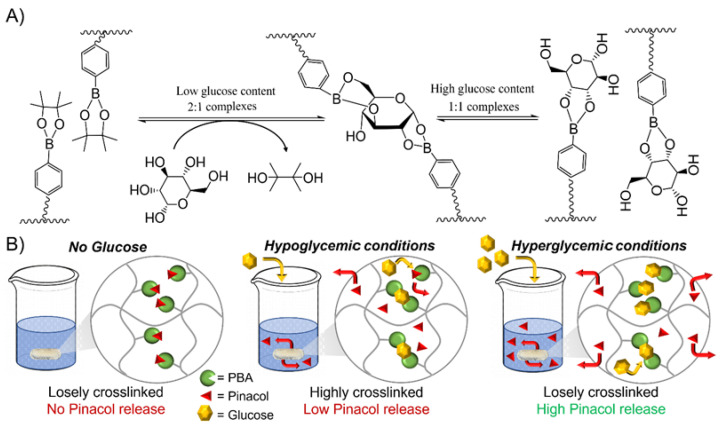
(**A**) Schematic representation of interaction of phenylboronic acid pinacol ester with glucose at varying glucose levels. (**B**) Graphic representation of the reversible molecular assemblies capable of trapping and releasing insulin by transesterification reaction at varying glucose concentrations, converting the 2:1 complex into two 1:1 complexes, which in turn decrease the number of crosslinks and engender an expansion in the hydrogel volume to accelerate the drug release kinetics.

**Figure 3 polymers-16-02502-f003:**
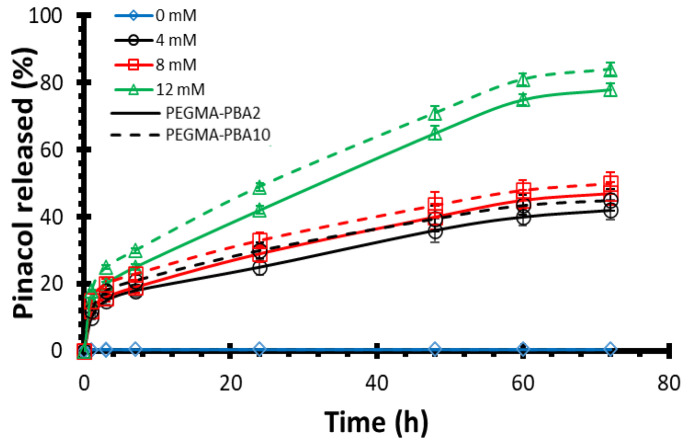
Pinacol release kinetics profiles of PHEMA-PBA2 (solid line) and PHEMA-PBA10 (dashed line) at varying glucose levels (i.e., 0 mM (blue), 4 mM (black), 8 mM (red) and 12 mM (green)).

**Figure 4 polymers-16-02502-f004:**
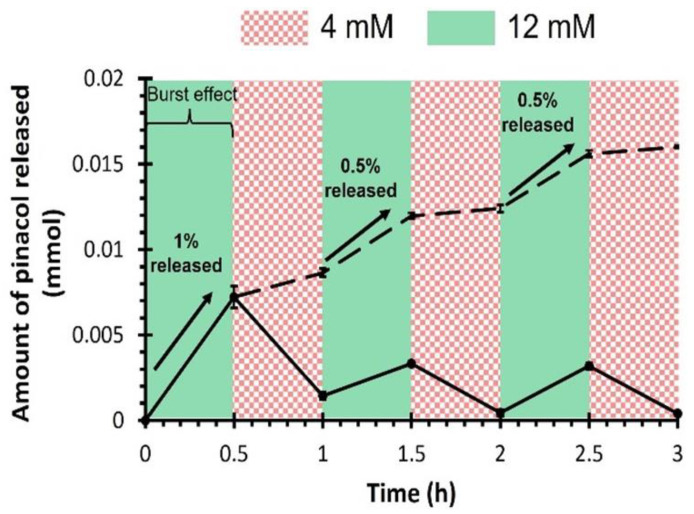
Pulsatile release of pinacol of PHEMA-PBA10 by alternately immersing in hypoglycemic and hyperglycemic conditions (i.e., 4 mM and 12 mM), leading to the amount of pinacol released at each cycle step (full line) and the total amount of pinacol released over time (dashed line).

**Figure 5 polymers-16-02502-f005:**
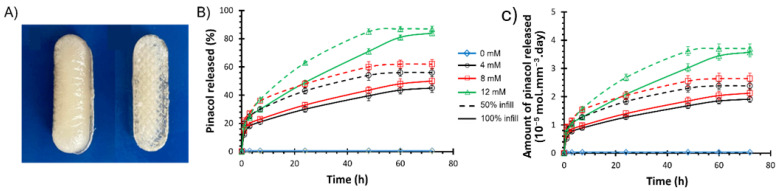
(**A**) Visual aspect of 3D-printed drug eluting implants with an infill density of 100% (left) and 50% (right). (**B**) Relative release of pinacol and (**C**) amount of pinacol released rationalized by the volume of the implant for the PHEMA-PBA10-based drug eluting implants with an infill density of 100% (solid line) and 50% (dashed line) at varying glucose levels (i.e., 0 mM (blue), 4 mM (black), 8 mM (red) and 12 mM (green)).

## Data Availability

The authors confirm that the data supporting the findings of this study are available within the article and its Appendix A. Further inquiries can be addressed to the corresponding author.

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
