# Peer review of "3D-Printed Phenylboronic Acid-Bearing Hydrogels for Glucose-Triggered Drug Release"

_polymers, 2024, doi:10.3390/polym16172502_

Round 1

Reviewer 1 Report

Comments and Suggestions for Authors

Please refer the document attached for detailed comments.

Comments on the Quality of English Language

The quality of English language is good. The sentences are short and concise with good clarity .

Author Response

Dear Referee,

Thank you for your encouraging message regarding our manuscript entitled “3D-Printed Phenylboronic Acid-bearing Hydrogels for Glucose-Triggered Drug Release”. We would like to take the opportunity to thank you for the careful reading of the manuscript and for your constructive comments. The comments and suggested modifications have clarified and improved the manuscript. In the next section, we are addressing point-by-point your comments and summarize the corresponding changes made in the revised manuscript.

This paper provides a proof of concept about a 3D printed hydrogel system for glucose triggered release of drugs to treat diabetes. The functionality of their system is demonstrated by measuring the release of different concentrations of pinacol, a molecule with release profile like insulin, under normal, hypoglycemic, and hyperglycemic conditions. The authors have discussed current delivery mechanisms for treating diabetes and how advancements in the healthcare field can enhance the overall quality of life. Their introduction is very detailed, with proper referencing. While the illustration of their work is good, from an experimental point of view, a more in-depth study of the release profile with different print designs and concentrations of polymers could be added to demonstrate the versatility and tailorability of the system.

  • We would like to take the opportunity to thank the Reviewer for the careful reading of the manuscript and for the constructive comments. The suggested in-depth study of the release profile with different print designs and concentrations of polymers would be the scope of our next papers, focusing on on-demand drug-releasing implant from the generation of gradient-like 3D-printed structures where each individual building layer has different drug-eluting properties using high resolution stereolithography.

Abstract:

Line 2: It is not mentioned anywhere else in the manuscript that this delivery system being proposed is a subcutaneous implant. Authors should clarify on this in the main body if its mentioned in the abstract.

  • According to the reviewer’s comment, subcutaneous is removed.

Line 9: ‘….properties than insulin’ is not the right usage here.

  • Line 9 is accordingly replaced by : “As a proof of concept, varying glucose concentration from hypoglycemia to hyperglycemia levels readily allow the release of pinacol, e. a 1,2-diol-containing model molecule, at respectively low and high rates.”

Introduction:

1st Paragraph:

Line 1: …most worldwide diffuse chronic disease…’ the sentence needs to be restructured for better clarity.

  • Line 1 is accordingly replaced by : “ Diabetes is a chronic disease characterized by persistently high blood glucose levels, making it one of the most prevalent global causes of mortality and morbidity.

Line 4: References should be added here.

  • Reference 2 is added.

Line 6: use of ‘e.g’ is not required in that sentence.

  • ‘e.g’ is removed.

Line 13: Could you explain what you meant by ‘competitive set point’ here.

  • The sentence was not clear, so that the sentence is replaced by : “Therein, insulin-lectin (e.g. concanavalin A) complexes are designed to release insulin upon a competitive binding with the triggering glucose but still its competitive set-point is above the range of typical hyperglycaemic concentrations.”

Line 27: Wrong referencing: ref 18

  • We thank the reviewer for his comment and we apologize for the wrong referencing. Accordingly, references are updated.

Line 30: Reference formatting is not correct.

  • We apologize for the wrong referencing and accordingly updated the references.

Line 30-32: Please provide some references to support the claim that PBA has higher operational stability than glucose oxidase and lectin.

  • Although it is difficult to find references that compare PBA, glugose oxidase and lectin, the stability of PBA is discussed in Reference 12.

2nd Paragraph:

Line 1: The sentence in its current form lacks clarity and readability. Please rephrase the sentence.

  • Line 1 is accordingly replaced by : “Fast and stable bond formation between PBA and glucose enable reversible molecular assemblies capable of trapping and releasing insulin for the tight regulation of blood glucose levels.”

Experimental:

3D printing of bioinert phenylboronic acid-containing hydrogels

Authors should mention what the final print structure and dimensions are here.

  • According to the reviewer’s comment, the print structure and dimensions are added as follow: “capsules of 30 mm length and 10 mm diameter”.

Results and Discussion:

3D printing of bioinert phenylboronic acid-containing hydrogels:

Subheading is same as in the experimental section.

  • Subheading is voluntarily same as in the experimental section to correlate the data results with the fabrication procedure.

Paragraph 1, Line 7: ‘thereby’ is redundant in this sentence

  • “Thereby” is accordingly removed.

Paragraph 2, Line 10-15: ‘….only 31% of pinacol is released…..’ Authors should include how many samples sets were used and give standard deviation for the measurements.

  • According to the reviewer’s comment, we added the following in the experimental section : “Drug release testing are performed on 5 specimens with identical parameters to provide mean values and standard deviations.” Standard deviations are around 1%.

Drug release kinetics:

Line 16 and Fig 5: It is seen that there is an initial burst release of almost 15% during first hour. Please explain why you see such high burst release. Also comment on whether this initial burst release can cause any side effects in practical situation

  • Our drug-eluting implants are designed to release drug upon a competitive binding with the triggering glucose, so that glucose displaces pinacol that is bound to PBA depending on the glucose concentration, thus causing its release within the first hours. Although there is almost 15% of drug released during the first hours, our PBA-containing PHEMA-based hydrogels can be used as an on-off switching system as a function of the glucose concentration, enabling on-demand release of the desired molecule, which in turn prevent any harmful drug release when not required. Still, simply washing the final implant in glucose solution of normal-glycemia conditions would allow the implant to equilibrate the pinacol-glucose competition, thus avoiding such a high burst effect. The burst effect could be further temperate both by adding specific substituent on the PBA moiety, limiting the release at undesired moments.

Conclusion:

Line 19: 2 X 10-6 mol/h instead of 2 10-6 mol/h

  • We updated Line 19 as requested.

Line 21:’….allows to deliver 2 units of insulin….’ Insulin is not used in any of the experiments. Authors needs to show experimental backing for this claim with insulin.

  • We agree with the reviewer that insulin is not used in any of the experiments. To avoid confusion, we removed the following from the conclusion, so that the sentence “Such rate allows to deliver e.g. 2 units of insulin, corresponding to the required dose of insulin to treat a 4 mM increase in glucose concentration within 3 minutes.” is erased.

General:

There is already literature on glucose-triggered insulin systems based on PBA. Can authors clarify on the novelty of their work and whether their system shows more efficiency than previous systems. The figures need a figure heading. What will happen to the implant over time? Do you observe any degradation under biological condition.

  • According to the reviewer’s comment, we would like to summarize a few key aspects that we believe set this manuscript apart and are of interest to the wide spectrum of readership. Our contribution reports on creating new phenylboronic acid-bearing hydrogels towards the high resolution stereolithography 3D printing with controlled designs for applications in diabetic treatments. Our core technology relies on controlling the release of a drug of interest by means of variations in crosslinking density upon glucose concentration. To that end, a newly photocurable methacrylated-phenylboronic acid derivative is synthetized involving a two-step synthesis. While glucose responsiveness has been assessed to the derivative itself, changes in crosslinking density are observed upon glucose concentration in 3D-printed phenylboronic acid-bearing hydrogels. In addition, our resulting hydrogels allow controlling the release kinetics of a model molecule, pinacol, with respect of the glycemia level. Overall, a pulsatile release test assesses the responsiveness of the hydrogels towards glucose concentration. Finally, a control of the design toward stereolithography 3D printing allows the modification of the kinetic release as a function of the surface-volume ratio. We believe such investigation are not reported elsewhere and make the novelty of the present contribution. Regarding the implant, no degradation over time is reported so far.

Reviewer 2 Report

Comments and Suggestions for Authors

The manuscript titled “3D-Printed Phenylboronic Acid-bearing Hydrogels for Glucose-Triggered Drug Release” by Odent et al. reported the 3D printed implants prepared from new phenylboronic acid (PBA)-containing polyhydroxyethyl methacrylate (PHEMA)-based hydrogels which can be activated to release encapsulated pinacol molecules by fluctuating glucose concentrations. The authors stated that their system will use diol-functionalized insulin. The authors claims that the dosage and release behavior of encapsulated biomolecules can be tailored by simply adjusting the size and geometry of 3D-printed implants. Therefore, the manuscript addresses the potential interest of the study.

However, there is a need of clarification in provided data sets to support the conclusion, and thus it seems premature to proceed with the manuscript based on the current results. Hence, I recommend revision of the manuscript to refine and answer the following points.

1.The authors should add the %yield of diol-functionalized insulin chemical modification reaction. 

2. The carboxylic acid protons (–COOH) peak is not visible in NMR spectrum of compound 1 pinacol-protected phenylboronic acid.

3. The authors should add the reason behind specifically choosing 4mM, 8mM and 12mM glucose concentrations.

4. Figure 3 legend contains term PHEMA-PBA2 which has not been used anywhere else in the manuscript. The authors should comment on it.

5. The authors used 3D-printed drug eluting implants with a 50% and 100% infill density. What would be the amount of pinacol released in 75% or 80% infill density? The authors should comment on it.

Comments on the Quality of English Language

Minor editing of English language required

Author Response

Dear Referee,

Thank you for your encouraging message regarding our manuscript entitled “3D-Printed Phenylboronic Acid-bearing Hydrogels for Glucose-Triggered Drug Release”. We would like to take the opportunity to thank you for the careful reading of the manuscript and for your constructive comments. The comments and suggested modifications have clarified and improved the manuscript. In the next section, we are addressing point-by-point the comments and summarize the corresponding changes made in the revised manuscript.

The manuscript titled “3D-Printed Phenylboronic Acid-bearing Hydrogels for Glucose-Triggered Drug Release” by Odent et al. reported the 3D printed implants prepared from new phenylboronic acid (PBA)-containing polyhydroxyethyl methacrylate (PHEMA)-based hydrogels which can be activated to release encapsulated pinacol molecules by fluctuating glucose concentrations. The authors stated that their system will use diol-functionalized insulin. The authors claims that the dosage and release behavior of encapsulated biomolecules can be tailored by simply adjusting the size and geometry of 3D-printed implants. Therefore, the manuscript addresses the potential interest of the study. However, there is a need of clarification in provided data sets to support the conclusion, and thus it seems premature to proceed with the manuscript based on the current results. Hence, I recommend revision of the manuscript to refine and answer the following points.

 1.The authors should add the %yield of diol-functionalized insulin chemical modification reaction.

  • Diol-functionalized insulin chemical modification reaction and related yield are discussed in the following reference DOI 11023/a:1015992828666
  1. The carboxylic acid protons (–COOH) peak is not visible in NMR spectrum of compound 1 pinacol-protected phenylboronic acid.
  • The carboxylic acid proton should endow a very tiny and broad signal around 12 ppm. The signal being so small, we cannot see the signal using the scale used in the NMR spectrum of Figure S1. Accordingly, we added within the caption of Figure S1 the following: The acid proton resonance around 12 ppm is not shown.”
  1. The authors should add the reason behind specifically choosing 4mM, 8mM and 12mM glucose concentrations.
  • It is completely normal for your blood sugar levels to go up and down every day in response to the food you eat. Between around 60 and 140 milligrams of sugar per deciliter of blood (mg/dl) is considered to be normal in people who don’t have diabetes. This is equivalent to a blood sugar concentration of between 4 and 8 mmol/l. If someone has readings over 8 mmol/l (140 mg/dl), they are considered to have high blood sugar (hyperglycemia). People with blood sugar levels below 4 mmol/l (60 mg/dl) are considered to have low blood sugar (hypoglycemia). But, as you can see in the illustration below, there are no clear-cut borders between normal blood sugar levels and too high or too low blood sugar.
  1. Figure 3 legend contains term PHEMA-PBA2 which has not been used anywhere else in the manuscript. The authors should comment on it.
  • We thank the reviewer for the comment. All data of PHEMA-PBA2 can be found in the supporting information. We simply selected PHEMA-PBA10 as the main sample for the discussion within the manuscript.
  1. The authors used 3D-printed drug eluting implants with a 50% and 100% infill density. What would be the amount of pinacol released in 75% or 80% infill density? The authors should comment on it.
  • We thank the reviewer for the question. Infill density of 50% and 100% are typical densities, so that we designed implants based on 50% and 100% infill density. However, if a 78 or 80% infill density would be used, the release kinetics would simply be proportional since they are diffusive and we are controlling the surface area to volume ratio of the drug eluting implant.